# Modeling the Competition between Misfolded Aβ Conformers That Produce Distinct Types of Amyloid Pathology in Alzheimer’s Disease

**DOI:** 10.3390/biom12070886

**Published:** 2022-06-24

**Authors:** Guilian Xu, Susan Fromholt, David R. Borchelt

**Affiliations:** 1Department of Neuroscience, College of Medicine, University of Florida, Gainesville, FL 32610, USA; xugl@ufl.edu (G.X.); sfromhol@ufl.edu (S.F.); 2Center for Translational Research in Neurodegenerative Disease, College of Medicine, University of Florida, Gainesville, FL 32610, USA; 3McKnight Brain Institute, College of Medicine, University of Florida, Gainesville, FL 32610, USA

**Keywords:** Alzheimer’s disease, amyloid pathology, prion strains, transgenic mice

## Abstract

The amyloid pathology characteristic of Alzheimer’s disease (AD) can be broadly classified as either fibrillary amyloid or diffuse amyloid. Fibrillary amyloid is found in cored-neuritic deposits, fibrillar deposits, and vascular deposits, and binds strongly to the amyloid revealing dyes Thioflavin-S or Congo Red. Diffuse amyloid can appear as wispy dispersed deposits or compact tufted deposits dispersed in neuropil, and binds amyloid dyes weakly if at all. In AD brains, both types of pathology are detected. Homogenates from AD brains, or the brains of transgenic mice modeling AD-amyloidosis, have been used to seed pathology in vulnerable host transgenic models. These studies suggest that pathologies may arise from distinct conformers or strains of misfolded Aβ, similar to propagating prions. Using Aβ strains sourced from four different AD-amyloidosis models, we injected pathological seeds into the brains of newborn mice from three different transgenic hosts with distinctive Aβ pathologies. Two of the seeding sources were from mice that primarily develop cored-neuritic Aβ deposits (cored strain) while the other two seeding sources were from mice that develop diffuse Aβ deposits (diffuse strain). These seeds were injected into host APP mice in which the resident strain was either diffuse or cored-neuritic pathology. Seeding-homogenates were injected into the brains of newborn mice to initiate propagation as early as possible. Depending upon the level of transgene expression in the host, we show that the injected strains of misfolded Aβ from the seeding homogenate were able to outcompete the resident strain of the APP host model. In serial passaging experiments, it appeared that the diffuse strain was more easily propagated than the cored strain. Collectively, our studies align with the idea that different types of Aβ pathology in AD brains arise from different populations of Aβ conformers that compete to populate the brain.

## 1. Introduction

In most cases of age-related dementia, neuropathological examination reveals Alzheimer’s disease (AD) [1,2,3]. To be classified as AD, two primary pathological hallmarks must be present, namely abnormal extracellular deposits of β-amyloid (Aβ) peptide and intracellular neurofibrillary tau tangles (NFTs) [2,3]. In familial forms of early-onset AD (FAD) caused by mutations in the amyloid precursor protein (APP) or presenilin (PS) 1 and PS2, the deposition of Aβ peptides in amyloid plaques is hypothesized to trigger a pathological cascade that ultimately leads to NFT pathology and clinical dementia [4]. Furthermore, polymorphisms in apolipoprotein E (APOE) and clusterin (CLU) may increase the risk of AD by a mechanism involving modulation of Aβ deposition [5,6,7,8]. Together, these genetic studies indicate that the deposition of Aβ can be a powerful trigger in causing age-related dementia.

Neuropathologically, the various types of amyloid in human AD brains can be classified on the basis of staining with amyloid binding dyes, morphology, and location [9,10,11,12,13,14]. In neuropil, fibrillar plaques and dense-cored plaques are strongly Thioflavin-S (Thio-S) positive [9,10,11]. The fibrillar deposits have a central mass of aggregated Aβ with spoke-like extensions leading to a confluent outer rim, whereas the dense-cored plaques have a compact core of Aβ aggregates surrounded by a less dense perimeter of Aβ. Both of these structures typically contain abnormal neuronal projections, termed neurites, and are surrounded by reactive astrocytes and microglia. Fibrillar and dense core plaques are both also stained by Congo Red to produce birefringence under polarized light. Semi-quantitative assessment of the burden of cored-neuritic Aβ deposits in frontal cortices is one of the pathological criteria for classification as AD [12]. Diffuse Aβ deposits are less organized, weakly stained or negative for Thio-S and Congo Red. Diffuse amyloid is usually present to varying degrees in AD brains, and is also commonly found in brains of individuals that were cognitively normal at death [12,13,14]. Fibrillar and cored deposits can also be found in aged individuals that were cognitively normal at death, but the frequency is usually lower [12,13,14]. These distinct Aβ pathologies have been reproduced in transgenic mice that express APP genes with mutations linked to FAD [15]. The origins of these different types of Aβ pathology and their relationships to each other remain an unresolved question in AD.

Recent studies have used cryo-EM to examine the structure of amyloids purified from human AD brains, finding evidence for two main types of misfolded Aβ filaments [16]. Type I filaments were the predominant structure in sporadic AD patients, while Type II filaments predominated in the brains of FAD patients with APP or PS1 mutations. The same study also examined filaments purified from individuals with other neurodegenerative diseases including Frontotemporal dementia, Dementia with Lewy Bodies, Parkinson’s disease dementia, aging-related tau astrogliopathgy, and pathologic aging [16]. In these non-AD dementia cases, Type II filaments were the only type of structure identified. Similarly Type II structures were exclusively isolated from brains of the APP^NL-F^ mouse model of Alzheimer amyloidosis. Neuropathological evaluations of the brains used for these structural studies showed a high abundance of cored-neuritic Aβ deposits in AD cases relative to non-AD cases and APP^NL-F^ mice. These intriguing findings suggest that the Aβ peptides that comprise cored-neuritic Aβ deposits may be structurally distinct from Aβ peptides in other types of amyloid deposits.

Other evidence has emerged that the misfolded Aβ in AD pathology exists in multiple conformations from studies in which AD pathology was induced in mice by prion-like seeding approaches. Seminal studies by Kane et al. were among the first to demonstrate that Aβ pathology could be induced in young APP transgenic host mice by injection of small amounts of brain homogenates prepared from human AD brains or from older APP mice with high amyloid burden [17]. Further to these initial studies, many laboratories have since demonstrated that Aβ pathology can be seeded in a prion-like manner (for reviews see [18,19,20,21,22]). Data from these studies demonstrated that morphologically distinct types of Aβ pathology could be propagated by seeding, indicating that distinct Aβ conformers of misfolded Aβ populate different types of amyloid deposits [20].

Here, we used Aβ seeding methods to examine competition between Aβ conformers, or strains, that produce cored-neuritic deposits or diffuse amyloid pathology. We used four different APP transgenic mouse models to epitomize the diversity of Aβ pathology in humans [23,24,25] (Table 1). Two of the models exhibit primarily cored-neuritic deposits that are strongly Thio-S positive, and the other two models exhibit wispy or tufted deposits that are only nominally Thio-S positive. We used these four mouse models as sources of seeding homogenates that were injected into the brains of newborn host APP transgenic mice to examine strain competition between Aβ in seeding homogenates and Aβ produced inherently by the transgenic model. Our findings in this paradigm show that the injected seeds outcompeted the inherent resident strain of Aβ, such that models that normally develop cored-neuritic deposits were converted to diffuse deposits. Similarly, an APP model that inherently developed diffuse deposits was converted to cored-neuritic pathology by seeding with cored Aβ strains. Notably, however, the seeding efficiency of homogenates from mice with cored-neuritic deposits appeared to be lower than that of homogenates from mice with diffuse deposits. Diffuse strains persisted through serial passage in a cored-strain host, but cored strains were outcompeted by diffuse strains when passaged through a diffuse-strain host. These findings reveal the dynamic relationship between the two major types of Aβ neuropathology that co-exist to varying degrees in AD and normal aging brains and are differentially associated with cognitive decline.

## 2. Methods

### 2.1. Transgenic Animals

The four seed sources (donors) used in this study have been described in previous studies (Table 1). Three of the seed sources were from models that use the MoPrP.Xho vector to express both APP and PS1 transgenes. Two of these models were generated by co-injecting two transgenic constructs that encode mutant APP and mutant PS1. One of these APP/PS1 models expresses a mouse APP695 cDNA that has been modified to produce human Aβ peptides, and to encode mutations associated with early-onset FAD that increase Aβ production (m/hAPPswe) [23,24]. This model co-expresses human PS1 encoding the exon 9 deletion mutation (originally known as APPswe/PS1dE9 Line 85 mice designated hereafter as PrP.HuAβ/PS1) [23,24]. The predominant type of Aβ pathology in this model is classified as cored-neuritic deposits that begin to develop around 6 months of age, with deposition occurring throughout the cortex and hippocampus [15]. The other APP/PS1 model expresses mouse APP695 cDNA with FAD mutations to produce murine Aβ peptides along with human PS1dE9 (originally known as MoAPPswe/PS1dE9 Line D943 mice designated hereafter as PrP.MoAβ/PS1) [25]. These mice exhibit very densely packed cored deposits that are concentrated in the white matter of the corpus callosum and along hippocampal fissures [25]. This model also shows abundant vascular deposition, particularly in the meninges. Onset of pathology in this model is around 14 months of age, but the density of deposits remains relatively low even when aged to 24 months [25]. The third model was an APP model in which m/hAPPP695 cDNA with 3 FAD mutations (Swe/Ind; K595M/N596L, V617F) is expressed via the PrP vector (designated PrP.APPsi mice) [25]. This model exhibits mixed pathology in which diffuse Aβ deposits are much more abundant than cored-neuritic deposits [26]. Onset of pathology in this model is around 12–14 months of age, with pathology present throughout the cortex and hippocampus. Vascular deposition is also evident in the meninges of the cerebellum and cortex [26]. The fourth source of brain homogenate was a model in which murine APP695 cDNA with the Swe/Ind mutations is expressed via transgene vectors that utilize tetracycline-regulated promoters and require mating to CamKIIa-tTA mice (designated hereafter as Tet.MoAβ) [25]. 

Three strains of mice used as seed sources were also used as recipients of seeding injections (Table 1). Unfortunately, the Tet.MoAβ model could not be recovered from cryopreservation and thus we were limited to using a seeding preparation described previously [26]. The mice used as seeding recipients were maintained with a hybrid strain background of C3H/HeJ × C57BL/6J, following a breeding scheme in which transgene positive males were bred to F1 B6/C3 mice. All mice requiring genotyping were marked by tail tattoo and genotyped via PCR of tail DNA, following protocols similar to those previously described [23,24]. PrP.APPsi mice, which co-express GFP in skin, were genotyped by visualizing GFP expression by illumination and visualization with special filter goggles (BLS Ltd., Budapest, Hungary) [25]. All animals were housed up to five per cage with unlimited access to food and water with a 14-h light and 10-h dark cycle. 

### 2.2. Inoculum Preparation

The initial seeding homogenates from PrP.MoAβ/PS1, Tet.MoAβ, HuAβ/PS1, and PrP.APPsi mice were the same as recently described [26]. These donor brains were from mice aged 24–27 months. Briefly, for the first passage, frozen mouse hemi-brain was sonicated in 10 volumes of sterile PBS (weight/volume), then the homogenate was clarified by centrifugation at 3000× *g* for 5 min. The supernatant was immediately aliquoted and stored at −80 °C until needed. For second passage homogenate, the brain was homogenized in six volumes of sterile PBS instead of 10×, due to lower levels of amyloid deposition. Every litter received freshly thawed aliquots that were discarded after a single day’s use.

### 2.3. Neonatal Intracerebral Seeding

Injections of the brain homogenates were performed as recently described [26]. Briefly, newborn pups (injected 20–40 h after birth, P0–P1) were anesthetized on ice, and 2 µL of inoculum was bilaterally injected into each cerebral ventricle using a 10 µL Hamilton syringe with a 30-gauge, 0.75 inch needle (Hamilton Company, Reno, NV, USA). Pups were placed on a heating pad for recovery for about 10–20 min and returned to the home cage.

### 2.4. Tissue Collection

Mice were anesthetized with isoflurane and perfused transcardially with 20 mL of cold PBS. The brains were quickly removed and cut sagittally through the midline, and one hemi-brain was immersion-fixed in 4% paraformaldehyde in PBS (pH 7.5) for ~48 h at 4 °C followed by processing and paraffin embedding. The other hemi-brain was frozen on dry ice and then stored at −80 °C. 

### 2.5. Histology and Immunochemistry

Paraffin sections (5 μm) were used for the histology and immunochemistry studies. The methods used in Campbell-Switzer (C-S) silver staining, ThioS staining, and standard immunochemistry protocols have been described previously [25]. The primary antibody used for Aβ immunostaining was a biotinylated monoclonal antibody (1:1000) termed MM27-33.1.1, provided by Dr. Todd Golde [27]. Antibody binding was detected using a ABC-horseradish peroxidase staining kit (Vector Laboratories, Burlingame, CA, USA) and then visualized by 3,3′-diaminobenzidine oxidation (KPL DAB reagent set; Seracare, Milford, MA, USA). Images of tissue sections were obtained using an Olympus BX60 microscope or by scanning with an Aperio^®^ XT System (Leica Biosystems, Buffalo Grove, IL, USA). The images were manipulated and cropped in GIMP (GNU Image Manipulation Program: www.gnu.org) and scale bars were added later in GIMP using pixel measurements.

## 3. Results

Our laboratory has generated and characterized multiple lines of transgenic mice expressing murine or humanized APP with various FAD mutations, with or without co-expression of mutant PS1 [23,24,25]. These various lines display diverse types of amyloid pathology ranging from primarily diffuse to primarily cored-neuritic deposits, with each having a distinct age to onset (Table 1). In a recent study, we established a novel seeding paradigm of Aβ pathology in APP transgenic hosts in which the seeding homogenates were injected into newborn mice [26]. The rationale behind this approach was to introduce the seeds as early as possible to override the inherent pathology of the model and re-program the system to produce the amyloid pathology enciphered by the injected seeds. By this approach, we demonstrated that a line of APP transgenic mice prone to develop diffuse amyloid deposits later in life (>12 months), termed PrP.APPsi mice, could be induced to develop cored deposits much earlier, by the injection of seeding homogenates from the brains of mice that primarily develop cored-neuritic deposits. This finding suggests that an Aβ strain prone to form cored deposits could replace the inherent diffuse strain of the host model. In the present study, we extended this line of investigation to examine competition between Aβ pathology induced by injection of seeds and the type of pathology inherent to the transgenic APP host mice. 

We used seeding homogenates from four different lines of APP transgenic mice that we recently described in seeding studies [26]. Briefly, we used as seed sources brains from two mice that developed primarily diffuse (Df) Aβ pathology (PrP.APPsi and Tet.MoAβ) and from two mice that developed cored-neuritic (Cr) Aβ pathology (PrP.MoAβ/PS1 and PrP.HuAβ/PS1) (Table 1; Figure 1). Examples of the morphology of Aβ deposits in these mice revealed by Campbell-Switzer silver (C-S silver) staining and Thio-S staining are shown at the top of Figure 1 and Appendix A. In both the PrP.APPsi (Df) and Tet.MoAβ (Df) mice, the Aβ pathology revealed by C-S silver staining appeared as wispy diffuse deposits (Figure 1) that were Thio-S negative (Appendix A). In the PrP.HuAβ/PS1 (Cr) and PrP.MoAβ/PS1 (Cr) mice, the Aβ pathology revealed by C-S staining was dominated by compact cored deposits (Figure 1) that were also strongly stained by Thio-S (Appendix A).

The recipients for these seeds were newborn mice from the PrP.MoAβ/PS1 (Cr) and PrP.HuAβ/PS1 (Cr) lines that were subsequently aged 9–12 or 6 months, respectively, before analysis by C-S silver staining (Figure 1a–j), Thio-S staining (Appendix A), and Aβ immunostaining (Appendix A). The PrP.MoAβ/PS1 mice normally develop pathology well after 12 months of age (Table 1; Figure 1a) [25]; hereafter we refer to these mice as late-depositing. Any deposits present in 12-month-old PrP.MoAβ/PS1 mice that had been injected with seeding homogenate would be interpreted as a product of seeding. Injecting newborn late-depositing PrP.MoAβ/PS1 (Cr) mice with seeds from PrP.APPsi (Df) or Tet.MoAβ (Df) mice resulted in robust Aβ deposition, rated as ++ to +++, by 9–12 months of age (Table 2; Appendix A). The deposits were diffuse, often wispy (Figure 1b,c,b’–c”; Appendix A), and largely Thio-S negative (Appendix A). Injecting newborn late-onset PrP.MoAβ/PS1 (Cr) mice with brain homogenates from mice that exhibit cored-neuritic deposits induced cored, Thio-S positive, deposits at 9–12 months of age; however, the overall burden of deposition was low (Figure 1d,e,d’–e”; Appendix A). These findings indicated that the diffuse strains present in the seeding homogenate from PrP.APPsi (Df) and Tet.MoAβ (Df) mice could replace the cored-strains inherent in late-onset PrP.MoAβ/PS1 mice. 

We also injected these same PrP.APPsi (Df) and Tet.MoAβ (Df) seeding preparations into the brains of newborn PrP.HuAβ/PS1 mice, which would normally develop cored deposits by 6 months of age (Figure 1f; Appendix A) [24]; hereafter referred to as early-depositing. When we examined the brains of seeded PrP.HuAβ/PS1 (Cr) mice, we largely observed the cored-neuritic pathology typical of this model (Table 2, Figure 1f–g,g’–j”; Appendix A). The overall burden of deposition in the seeded PrP.HuAβ/PS1 mice was rated no higher than that of uninjected, aged-matched, PrP.HuAβ/PS1 mice (Table 2, Figure 1; Appendix A), except when seeds from Tet.MoAβ mice were injected. In this case, we observed a modest increase in the level of diffuse Aβ pathology, leading to a mixed pathology (M), rated as ++ in severity by a blinded observer (Table 2, Figure 1h). Importantly, in all the lines of host mice used, injection of seeding homogenates into newborn nontransgenic (NTg) littermates produced no pathology or obvious abnormality (data not shown). From this set of seeding experiments, we observed a relationship between the type of Aβ pathology exhibited by the seed source, the type of pathology endemic to the recipients of the seeds, and the normal age that recipients develop pathology without seeding. The PrP.HuAβ/PS1 mice (6 mo onset inherently) were less influenced by seeding, indicating that the seeded strain could not completely outcompete the inherent strain of the host. The seeds prepared from PrP.APPsi and Tet.MoAβ mice were highly potent in late-depositing PrP.MoAβ/PS1 mice, and induced primarily diffuse Aβ pathology similar to the seed source. Overall, the data are consistent with the idea that the different types of pathology are caused by distinct strains of misfolded Aβ that compete for dominance. 

To determine whether the Aβ strain seeds could retain their original characteristics after passage through a host, we prepared homogenates from first-passage recipients to inject into naïve hosts. The sequence of injections was as follows. Homogenate from Tet.MoAβ mice (Figure 2a) was first injected into the brains of newborn PrP.APPsi (Df) mice and PrP.MoAβ/PS1 (Cr) mice. From each of these first-passage recipients, we selected a representative animal to use in preparation of new seeding homogenates that were then injected into newborn mice from three different transgenic APP recipients; early-depositing PrP.HuAβ/PS1 (Cr) mice, late-depositing PrP.APPsi (Df) mice, and late-depositing PrP.MoAβ/PS1 (Cr) mice (Table 3; Appendix A). When we passaged the diffuse Aβ strain of Tet.MoAβ mice through PrP.APPsi (Df) mice into the same 3 recipient hosts, the recipients of these second passage seeds produced diffuse Aβ pathology, as expected (Figure 2a–e,b’–e”). In parallel, we passaged the diffuse strain from Tet.MoAβ mice (Figure 3a) through PrP.MoAβ/PS1 (Cr) mice (Figure 3b,b’,b”) into the same 3 recipients. When the early-depositing PrP.HuAβ/PS1 mice was seeded with second-passage diffuse strain Aβ, we observed a modest increase in diffuse Aβ pathology (Figure 3c,c’,c”). In the two late-onset hosts, PrP.APPsi and PrP.MoAβ/PS1, secondary passaging induced a robust increase in diffuse Aβ deposition in both cases (Figure 3d,e,d’–e”). These findings suggest that the diffuse Aβ strain from Tet.MoAβ mice outcompeted the inherent cored-strain of late-depositing PrP.MoAβ/PS1 mice, such that seeds prepared from first-passage recipients retained the characteristics of the diffuse-strain.

To examine whether cored strains could survive passage through hosts that preferentially develop diffuse pathology, we passaged seeds from PrP.HuAβ/PS1 (Figure 4a–d,b’–d”) and PrP.MoAβ/PS1 (Figure 4e–h,f’–h”) mice through PrP.APPsi mice, and then back into either PrP.HuAβ/PS1 (Cr) or PrP.APPsi (Df) mice. The pathology burden in PrP.HuAβ/PS1 mice injected with second passage core-seeds from either primary source was similar to that of uninjected mice, and the predominant types of deposits were cored (Figure 4c,c’,c”,g,g’,g”). Notably, two PrP.HuAβ/PS1 mice that received second passage core-seeds showed increased levels of diffuse pathology (Table 3, Figure 4g), suggesting that the diffuse strain inherent to PrP.APPsi was not fully outcompeted by the core-strain seeds in the first passage. Additionally, when the first-passage core-strain seeds were injected into late-onset PrP.APPsi mice, we observed a robust induction of diffuse pathology (Figure 4d,d’,d”,h,h’,h”). These findings further reinforce the suggestion that the inherent diffuse strain of the first-passage PrP.APPsi recipient mice was not completely outcompeted and re-emerged as the dominant strain when introduced into a host that inherently favored diffuse pathology.

## 4. Discussion

Using a novel approach of injecting seeding homogenates into newborn mice, we examined Aβ strain competition and dominance in APP transgenic mice. We also extended studies of seeding interactions between human and murine species of Aβ that we originally reported recently [26]. Collectively, across these studies we found no appreciable species barrier between human and murine Aβ, in terms of seeding amyloid deposition in mice expressing mutant forms of human or murine APP. Human Aβ seeds were highly efficient in seeding host transgenic mice expressing murine APP transgenes. Similarly, murine Aβ seeds derived from transgenic mice were highly efficient in seeding host mice expressing humanized APP transgenes. We also demonstrated that in host mice that normally develop pathology later in life (>12 months of age) it was possible to override the host Aβ strain by injecting with homogenates from mice containing morphologically distinct types of Aβ pathology. Injection of newborn late-depositing PrP.APPsi mice (Df) with seeding homogenates from the brain of an older PrP.HuAβ/PS1 mouse (Cr) produced mice that predominantly exhibited cored deposits at 12 months of age, albeit at modest levels. Injection of newborn late-depositing PrP.MoAβ/PS1 mice (Cr) with brain homogenates from older PrP.APPsi or Tet.MoAβ mice (Df) produced mice with robust levels of diffuse Aβ pathology by 12 months post-injection. Additionally, the diffuse strain of Aβ pathology could be passaged serially from seeded PrP.MoAβ/PS1 (Cr) mice to naïve newborn PrP.MoAβ/PS1 (Cr) mice. In early-onset PrP.HuAβ/PS1 (Cr) mice, we were able to increase modestly the level of diffuse pathology by injection of homogenates from Tet.MoAβ mice, but in all other cohorts the injected PrP.HuAβ/PS1 mice had levels of cored Aβ deposits at the time of harvest that were similar to uninjected, age-matched, mice. The weak response of early-depositing PrP.HuAβ/PS1 mice to seeding suggests that the cored strain of Aβ inherent to this model is produced at a sufficient level to outcompete the introduced diffuse strain. Collectively, our findings are consistent with the idea that the diffuse and cored amyloid pathology in these models is generated by distinct strains of misfolded Aβ that compete to populate the brains of susceptible hosts.

Notably, studies that have examined potential cases of iatrogenic transmission of Aβ pathology in humans have reported increased incidence of diffuse and vascular deposits in suspected cases [28,29,30,31,32,33,34,35,36]. To varying degrees, depending in part upon the route of exposure, these cases may demonstrate typical cored Aβ pathology [29,37,38,39,40]. In some of these cases, the transmitted vascular pathology may have contributed to clinical disease, but otherwise an association of these transmitted pathologies to AD-related cognitive impairment has been rare [37,41].

In most of the studies that have used human brains to seed APP transgenic mice, the induced pathology in the seeding recipient has been described as diffuse (reviewed in [21]). In a recent study in which we seeded newborn PrP.APPsi mice with human AD brain homogenates, we also observed predominantly diffuse Aβ deposition [26]. The brains of AD patients frequently simultaneously exhibit cored and diffuse types of Aβ pathology. Cross-sectional neuropathological studies of aged humans have suggested that diffuse Aβ deposits are the first pathology to appear, followed by cored deposits becoming more prevalent with age or declining cognitive status [2]. These findings have advanced the idea that there is a progressive relationship between the two types of pathology, with diffuse Aβ deposits maturing with age into cored-neuritic deposits. Our seeding data suggest that these two types of Aβ-deposit pathology are truly distinct and competing, and that the appearance of maturation may instead reflect a delayed infiltration by cored strains of Aβ pathology over time. 

A recent study suggests that cored deposits may originate from Aβ aggregates that initially form within endosomal/lysosomal compartments [42]. It is not known whether diffuse deposits could also originate in intracellular compartments. In the late-depositing PrP.MoAβ/PS1 model, injected diffuse Aβ seeds effectively outcompeted the cored strain of misfolded Aβ. This finding suggests that some of the diffuse seeds might have infiltrated endosomal/lysosomal compartments. Alternatively, the diffuse Aβ seeds could have initiated Aβ fibrilization in extracellular compartments, effectively starving any cored-strain aggregates of Aβ peptides needed for growth. 

The three models used as seeding recipients in these studies were all generated from the same genetic background and used the same promoter element to drive transgene vectors. The only variable between the models was whether mutant PS1 was co-expressed with APP. In all three models, the mutations in the APP genes that promote higher production of Aβ42 peptides are located outside the boundaries of processed Aβ; and thus, we expected that the PrP.APPsi and PrPHuAβ/PS1 models should produce human Aβ peptides of the same sequence. The PrP.MoAβ/PS1 mice would produce murine Aβ peptides. Therefore, in these three models, the morphology of the Aβ deposits, and hence the strain of Aβ peptide produced, appears to be due to the influence of mutant PS1 on the processing of mutant APP. Given that APP processing to produce Aβ peptides is a proteolytic process [43], it is possible that the mice expressing mutant PS1 produced different mixtures of Aβ peptides [44], with the composition of these mixtures influencing early strain selection events in the competition between diffuse and cored strains of misfolded Aβ. Notably, in both the Tet.MoAβ (Df) and PrP.APPsi (Df) mice, the level of insoluble Aβ42 in formic acid fractions is 5–10 times higher than the level of Aβ40 [15]. In the PrP.HuAβ/PS1 (Cr) model, the levels of Aβ42 and Aβ40 in formic acid fractions are more similar [15]. In the PrP.MoAβ/PS1 (Cr) model, the level of formic acid extractable Aβ42 is about 3-fold higher than Aβ40 [15]. These findings are consistent with the idea that early strain selection may involve the assembly of mixtures of Aβ peptides, producing misfolded conformers that ultimately selectively propagate through assembly of Aβ42 and, or Aβ40 peptides.

## 5. Conclusions

Our findings are consistent with the idea that different types of Aβ pathology are generated by distinct populations of misfolded Aβ, akin to prion strains. Competition between these strains appears to modulate the type of Aβ pathology that ultimately populates and spreads in an affected brain. In general, we noted that seeds prepared from mice with diffuse Aβ pathology appeared to be more potent; meaning that the burden of induced pathology was generally greater than when seeds were prepared from mice exhibiting diffuse Aβ pathology. We recognize that in terms of independent biological replicates our study is limited in its scope, and may not capture the full spectrum of variability. There were no obvious male/female differences in pathology among the seeded mice, but our study was not designed detect such differences. Future studies that involve larger cohorts of mice will be useful for determining whether the sex of the host may influence the induction of Aβ pathology by the seeding and propagation of Aβ strains. Overall, our data provide striking examples of competition between different strains of misfolded Aβ producing distinct types of pathological features. A re-examination of human AD pathology in which different types of Aβ are viewed as competitors may help resolve the extent to which the type of competition modeled here in mice drives the evolution of Aβ deposition in humans.

## Figures and Tables

**Figure 1 biomolecules-12-00886-f001:**
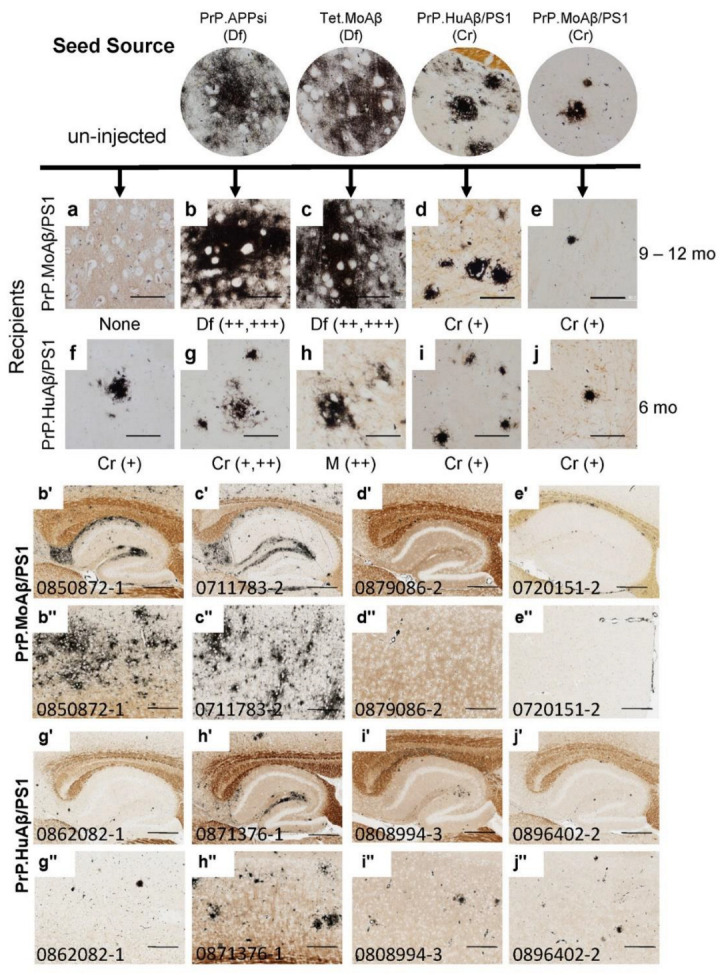
Competition between diffuse and cored strains of Aβ pathology in APP transgenic hosts. The upper panels show high-power images of Aβ deposit morphology, revealed by C-S silver staining (see Methods), in the brains of mice that provided seed sources [26] and the recipients of these seeding injections. The lower panels show low-power views of the seeded animals. Details of animal numbers for each injection are summarized in Table 2, with details of individual animals provided in Appendix A. PrP.MoAβ/PS1 mice were aged 9–12 months before harvest (**a**–**e**,**b’**–**e’**,**b”**–**e”**), and PrP.HuAβ/PS1 mice were aged 6 months post-injection (**f**–**j**,**g’**–**j’**,**g”**–**j”**). In the upper panel, below each row of images from the seeded animals, we provide information on pathological rating (Df = diffuse; Cr = cored) and relative severity. Blinded observers reviewed the images and scored the burden of Aβ pathology based on the following criteria: “+++” = heavy Aβ burden distributed throughout the cortex and hippocampus (examples are panels (**b’**,**c’**,**b”**,**c”**)). “++” = obvious pathology with patchy deposition in cortex and hippocampus (examples are panels (**h’**,**h”**)). “+” = consistent but minimal pathology (**d’**,**e’**,**g’**,**i’**,**j’**). (**a**–**j**) Images of amyloid deposits in cortex, imaged by microscope with 40× objective lens, scale bar = 50 µm. (**b’**–**j’**) Images of hippocampus cropped from 20× Aperio system scanned images, 5000 × 3750 pixels, scale bar = 500 µm. (**b”**–**j”**) Images of cortex cropped from 20× Aperio system scanned images, 2000 × 1500 pixels, scale bar = 200 µm. Each animal in the colony was given a number that was used to track identification. These numbers appear on images in each figure that follows.

**Figure 2 biomolecules-12-00886-f002:**
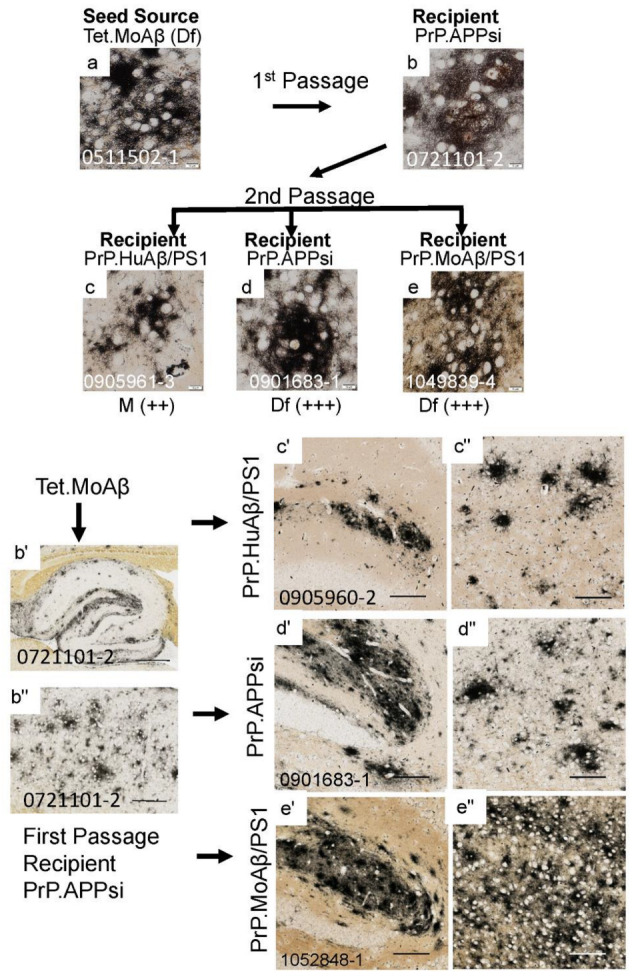
Serial passage of diffuse Aβ pathology through a PrP.APPsi host. The upper panels show high-power images of Aβ deposit morphology from C-S silver staining in the brain of the seed source animal, the brain of the first passage recipients, and the brains of the second passage recipients. Aβ pathology was confirmed by Aβ immunostaining (Appendix A). The scoring criteria for pathology are described in the legend to Figure 1 The lower panels show low-power views of the seeded animals. Details of animal numbers for each injection are provided in Table 3 and Appendix A. (**a**) Image from the cortex of the original seed source Tet.MoAβ animal [26]. (**b**,**b’**,**b”**) Images from the recipient PrP.APPsi mouse used as host for Passage 1 (aged 12 months before harvest). (**c**,**c’**,**c’**’–**e**,**e’**,**e’**’). Representative images of brain from recipients seeded with first passage brain from PrP.APPsi mice (**b**,**b’**,**b”**). In the upper panel, below each row of images from the second passage seeded animals, there is information on pathological rating (Df = diffuse; Cr = cored) and relative severity, using the same scoring system as in Figure 1. (**a**–**e**) Images of cortex. (**b’**–**e’**) Images of hippocampus. (**b”**–**e”**) Images of cortex. (**a**–**e**) Images of amyloid plaques in cortex taken by microscope with 40× objective lens, scale bar = 10 µm. (**b’**–**e”**) Images of hippocampus or cortex cropped by Aperio system scanned images (20×). (**b”**) Image of hippocampus, scale bar = 500 µm. (**b”**) Image of cortex, scale bar = 200 µm. (**c’**,**d’**,**e’**) images of hippocampus and (**c”**,**d”**,**e”**) images of cortex, cropped at 1000 × 1000 pixels, scale bar = 100 µm.

**Figure 3 biomolecules-12-00886-f003:**
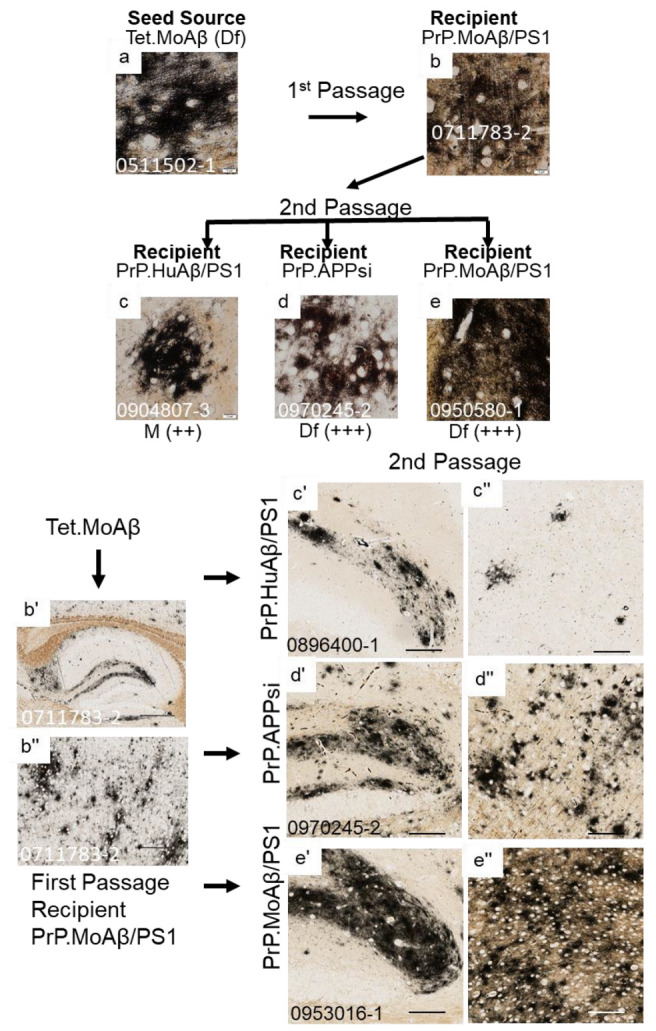
Serial passage of diffuse Aβ pathology through a PrP.MoAβ/PS1 host. The upper panels show high-power images of Aβ deposit morphology from C-S silver staining in the brain of the seed source animal, the brain of the first passage recipients, and the brains of the second passage recipients. Aβ pathology was confirmed by Aβ immunostaining (Appendix A). The criteria for scoring pathology burden were the same as described in the legend of Figure 1. The lower panels show low-power views of the seeded animals. Details of animal numbers for each injection are provided in Table 3 and Appendix A. (**a**) Image from the cortex of the original seed source Tet.MoAβ animal [26]. (**b**,**b’**,**b”**) Images from the recipient PrP.MoAβ/PS1 mouse used as host for Passage 1 (aged 12 months before harvest). (**c**,**c’**,**c”**–**e**,**e’**,**e”**). Representative images of brains from recipients seeded with first passage brain from PrP.MoAβ/PS1 mice (**b**,**b’**,**b”**). In the upper panel, below each row of images from the secondpassage seeded animals, information is provided on pathological rating (Df = diffuse; Cr = cored) and relative severity, using the same scoring system as in Figure 1. (**a**–**e**) Images of cortex. (**b’**–**e’**) Images of hippocampus. (**b”**–**e”**) Images of cortex. The scale bars are as shown in the legend of Figure 2.

**Figure 4 biomolecules-12-00886-f004:**
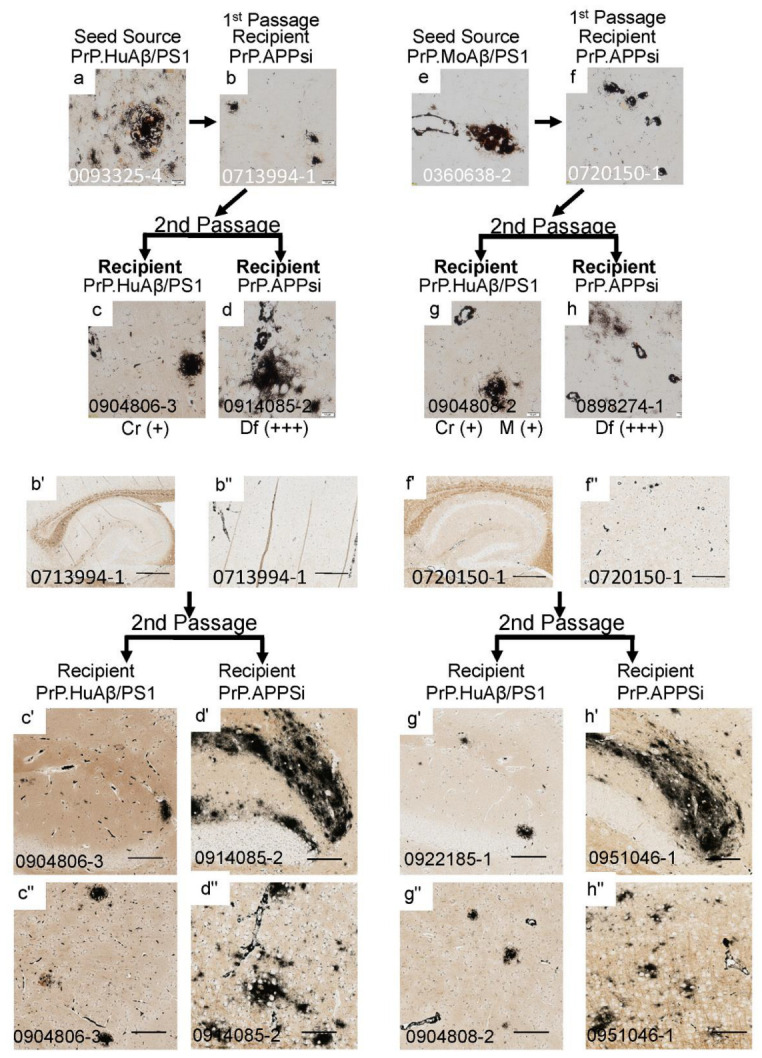
Serial passage of cored Aβ pathology through PrP.APPsi hosts. The upper panels show high-power images of Aβ deposit morphology from C-S silver staining in the brain of the seed source animal, the first passage recipients, and the second passage recipients. Aβ pathology was confirmed by Aβ immunostaining (Appendix A). The criteria for scoring pathology burden were the same as described in the legend of Figure 1. The lower panels show low power views of the seeded animals. Details of animal numbers for each injection are provided in Table 3 and Appendix A. (**a**) Image from the cortex of the original seed source PrP.HuAβ/PS1 animal. (**b**,**b’**,**b”**) Images from the recipient PrP.APPsi mouse used as host for Passage 1 (aged 12 months before harvest). (**c**,**c’**,**c”**,**d**,**d’**,**d”**) Representative images of brain from recipients seeded with first passage brain from PrP.APPsi mice. (**e**) Image from the cortex of the original seed source PrP.MoAβ/PS1 animal. (**f**,**f’**,**f”**) Images from the recipient PrP.APPsi mouse used as host for Passage 1 (aged 12 months before harvest). (**g**,**g’**,**g”**,**h**,**h’**,**h”**) Representative images of brain from recipients seeded with first passage brain from PrP.APPsi mice. In the upper panel, below each row of images from the 2nd passage seeded animals, there is information on pathological rating (Df = diffuse; Cr = cored) and relative severity, using the same scoring system as in Figure 1. (**a**–**h**) Images of cortex. (**b’**,**c’**,**d’**,**f’**,**g’**,**h’**) Images of hippocampus. (**b”**,**c”**,**d”**,**f”**,**g”**,**h”**) Images of cortex. The scale bars are as described in the legend of Figure 2.

**Table 1 biomolecules-12-00886-t001:** List of the mouse models used in this study.

Model Designation	Aβ Species	Transgenes	Onset Aβ Deposition (mo)	Neuropath Features of Aged Animals	Role in Study	Reference
**PrP.HuAβ/PS1**	Human	APPswe, PS1dE9	~6	Cored > > diffuse, Thio+ with CAA	Host/ Donor	[23,24]
**PrP.MoAβ/PS1**	Mouse	APPswe, PS1dE9	~14	Cored, Thio-S+ with CAA	Host/ Donor	[25]
**PrP.APPsi**	Human	APPswe/ind, GFP	~12	Diffuse > > Cored, Weakly Thio-S+	Host/ Donor	[25]
**Tet.MoAβ**	Mouse	APPswe, tTA, GFP	~13	Diffuse, mainly Thio-S neg	Donor	[25]

**Table 2 biomolecules-12-00886-t002:** First passage cohort data and results; Cr-core plaques, Df-diffuse plaques, M-mixed.

First Passage to PrP.MoAβ/PS1(Cr) Mice Harvested at 9–12 Months of Age			
Number of Mice Injected	Seed Source	Recipient Pathology Type at Harvest	Pathology Burden Score ^a^
5	None	None	-
3	PrP.APPsi (Df)	Df	++ to +++
4	Tet.MoAβ (Df)	Df	++ to +++
5	PrP.HuAβ/PS1 (Cr)	Cr	+ *
6	PrP.MoAβ/PS1 (Cr)	Cr	+ **
**First Passage to PrP.HuAβ/PS1(Cr) Mice Harvested at 6 Months of Age**			
3	None	Cr	+
4	PrP.APPsi (Df)	Cr	+ to ++
5	Tet.MoAβ (Df)	M	++
9	PrP.HuAβ/PS1 (Cr)	Cr	+
7	PrP.MoAβ/PS1 (Cr)	Cr	+

^a^—see legend to Figure 1 for an explanation of the scoring criteria. *—one mouse exhibited mixed pathology. **—one mouse lacked pathology.

**Table 3 biomolecules-12-00886-t003:** Summary of serial passage cohort data results; Cr-core plaques, Df-diffuse plaques, M-mixed.

Number of Mice Injected	Passage History (Host Strain)	Last Recipient Pathology Type at Harvest	Pathology Burden Score ^a^
6	Tet.MoAβ to PrP.APPsi to PrPHuAβ/PS1 **(Df to Df to Cr)**	M	++
3	Tet.MoAβ to PrP.APPsi to PrP.APPsi **(Df to Df to Df)**	DF	+++
5	Tet.MoAβ to PrP.APPsi to PrP.MoAβ/PS1 **(Df to Df to Cr)**	Df	+++
5	Tet.MoAβ to PrP.MoAβ/PS1 to PrP.HuAβ/PS1 **(Df to Cr to Cr)**	M	++
6	Tet.MoAβ to PrP.MoAβ/PS1 to PrPAPPsi **(Df to Cr to Df)**	Df	+++
3	Tet.MoAβ to PrP.MoAβ/PS1 to PrP.MoAβ/PS1 **(Df to Cr to Cr)**	Df	+++
6	PrP.HuAβ/PS1 to PrP.APPsi to PrP.HuAβ/PS1 **(Cr to Df to Cr)**	Cr	+
5	PrP.HuAβ/PS1 to PrP.APPsi to PrP.APPsi **(Cr to Df to Df)**	Df	+++
6	PrP.MoAβ/PS1 to PrP.APPsi to PrP.HuAβ/PS1 **(Cr to Df to Cr)**	Cr (4), M (2)	+
3	PrP.MoAβ/PS1 to PrP.APPsi to PrP.APPsi **(Cr to Df to Df)**	Df	+++

a—see legend to Figure 1 for an explanation of the scoring criteria.

## Data Availability

All relevant data is included in the manuscript.

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
