# Peer review of "Modeling the Competition between Misfolded Aβ Conformers That Produce Distinct Types of Amyloid Pathology in Alzheimer’s Disease"

_biomolecules, 2022, doi:10.3390/biom12070886_

Round 1

Reviewer 1 Report

The authors investigated the transmission competition of two distinct types of amyloid aggregated cored amyloid and diffused amyloid in vivo using transgenic mice model. They found that both of two types aggregates may seed the host mice and the diffused seed seems to be more effective. Overall, this work provides new insights on the cross-seeding phenomenon in amyloid and in my opinion this data presented in this work is publishable. Yet, the authors need to consider the following suggestions for revision.

(1)     The readers of this journal may not be in the field of medicine. The authors should provide more interpretations on the actual structures and properties of the two distinct types of amyloids, cored and diffused, from a biophysical viewpoint.  They should also cite more related references by structural biologists regarding to the structures of these amyloids by cryo-EM, XRD, and solid state NMR if these references are available.

(2)     Please provide more justifications on why the seeding experiments are useful in the treatment of AD. AD, unlike mad cow disease, it is not contagious. It cannot pass from one patient to another. In reality, how can seeding happen? If the seeding experiment is helpful for diagnostics, please provide more justifications.   

Author Response

Dear Editor,

We thank the reviewers for their constructive comments on our manuscript. We have made several revisions to the document and figures to address these comments. The changes have been marked or tracked.  Substantial changes include additional text in the Introduction.  All 4 primary figures were changed to improve the labeling. Tables 2 and 3 were replaced with new versions that are formatted differently and modified to include information on scoring requested by reviewer 2.  We have also modified the Supplementary Materials to include additional information on pathology scoring. 

We believe the manuscript has substantially improved and hope that it is now acceptable for publication.

David R. Borchelt

Response to Reviewer 1

(Comment 1)     The readers of this journal may not be in the field of medicine. The authors should provide more interpretations on the actual structures and properties of the two distinct types of amyloids, cored and diffused, from a biophysical viewpoint.  They should also cite more related references by structural biologists regarding to the structures of these amyloids by cryo-EM, XRD, and solid state NMR if these references are available.

Response: We thank the reviewer for this suggestion.  There is a recent study on the structure of amyloid isolated from human AD brain and other dementias that is very relevant and we have now discussed this paper in the Introduction.

(2)     Please provide more justifications on why the seeding experiments are useful in the treatment of AD. AD, unlike mad cow disease, it is not contagious. It cannot pass from one patient to another. In reality, how can seeding happen? If the seeding experiment is helpful for diagnostics, please provide more justifications.  

Response: We are using the seeding approach as a tool to study different types of amyloid pathology and how they are related to each other. We have made a few changes to the introduction to make this point clearer.  Seeding would not necessarily be useful in treatment and we have not advocated for using seeding for this purpose. 

Author Response

Dear Editor,

We thank the reviewers for their constructive comments on our manuscript. We have made several revisions to the document and figures to address these comments. The changes have been marked or tracked.  Substantial changes include additional text in the Introduction.  All 4 primary figures were changed to improve the labeling. Tables 2 and 3 were replaced with new versions that are formatted differently and modified to include information on scoring requested by reviewer 2.  We have also modified the Supplementary Materials to include additional information on pathology scoring. 

We believe the manuscript has substantially improved and hope that it is now acceptable for publication.

David R. Borchelt

Response to Reviewer 2

Comment 1 [Major]: Quantitative Statistics. While reading the manuscript, the authors do a great job of outlining the details of the experiments, but there is a lack of quantitative statistics present throughout the results. I understand that the authors had graded the images based on a +/++/+++ score for >5, >30 and TMTC respectively, but it would help if the reader knew how many samples were reviewed, what the average number of deposits were. This will not just help the paper itself, but also future readers and researchers who are interested in comparing results or modelling trends of amyloid deposits. If this detail is already provided somewhere that I’ve missed, my apologies.

Response: We thank the reviewer for pushing us to be more transparent. We should have included all the data in the initial submission. We have now updated the supplemental materials to include the scoring for each animal.  We have also clarified the scoring criteria and defined specific examples.  The data are not really the type of data that are amenable to statistics. The scoring system was used here to assess whether the seeding injection had actually induced pathology beyond what would occur without injection. The included supplemental tables now more transparently demonstrate the consistency of the data.

Comment 2 [Minor] Figure Labelling. The contrast of the letters on some of the images is too low to make out, especially for accessibility purposes. If the authors could more clearly indicate, using Figure 3 as an example, b’’ and e’/e’’ so that they are darker, that would be great. Same as other figures where the same contrast issue happens.

Response: We have modified the figures so that it is easier to see the labels.

Comment 3 [Major] Future Research and Discussion. The authors present a thorough discussion of their results and summarize them well in the conclusions. However, it would be great to add in an extra few sentences to outline what further studies can be done to help widen the scope of the results, either on the molecular or cellular level. The authors do recognize their study is limited in scope and may not capture the full spectrum of variability, so suggestions should be made to help mitigate this or at least provide more clarity into how further studies can show different levels of competitive growth of Aβ strains leading to certain pathologies.

Response: We have added a few sentences to the Conclusion section to clarify how larger cohort sizes would help address whether there may be sex dependent differences in seeding. We have also tried to identify what types of human pathology studies might be useful to determine the role of strain competition in the evolution of human AD pathology.